# Linguistic Structures for Formal Semantics of Natural Languages

**Alexey Stukachev**
Sobolev Institute of Mathematics
Novosibirsk State University
Novosibirsk, Russia
aistu@math.nsc.ru

Tsylu Tsao
Novosibirsk State University
Novosibirsk, Russia
t.tsao2@alumni.nsu.ru

Ulyana Zaitseva
Novosibirsk State University
Novosibirsk, Russia
u.penzina@g.nsu.ru

## Abstract

We discuss a notion of linguistic structure suitable to formalise semantics of natural language sentences and texts. In particular, a method of comparison for linguistic structures is presented, based on well-known edit distance measure from graph theory. Also, we study quantifier scope ambiguity (the Scope Problem). Traditional Montague-based approaches often require generating multiple logical forms for a single sentence. To overcome this, we integrate Hilbert's epsilon-operator and Skolem choice functions.

## 1 Introduction

In many examples from mathematical and computational linguistics, interpretations of structures can be viewed either as explanations and formalizations (of practical issues), or as adaptations and applications (of theoretical issues). The study of these two types of interpretations is important for current and future research in explainable artificial intelligence. At present, the main problem is the lack of adequate mathematical and computational models for such studies.

Classical notions and techniques from model theory are used in mathematical linguistics since Montague seminal works, with precise formal representations of language structures, both syntactical and semantic. However, in modern computational linguistics machine learning and statistical methods generate linguistic data structures (datasets) of different type. In particular, these structures do not always satisfy the requirements of formal correctness.

We choose an example of formal mathematical definition of a linguistic data structure and introduce an operation of comparison for such structures, formalized in algebraic and model-theoretic terms.

A natural and important question for linguistic structures is a comparison of relevance or similarity. In distributional language models, cosine similarity is a method for measuring semantic similarity of words based on their vector representations. We discuss a method for measuring semantic similarity for texts and linguistic datasets, by generating corresponding graph structures and calculating the graph editing distance.

A significant challenge in formalizing natural language semantics is quantifier scope ambiguity (the Scope Problem). Traditional Montague-based approaches often require generating multiple logical forms for a single sentence. To overcome this, we integrate Hilbert's epsilon-operator and Skolem choice functions. This approach creates a single, flexible representation of a sentence's meaning. Instead of forcing the AI to choose one interpretation immediately, Skolem choice functions define the relationships between different words. It allows the system to combine different datasets while keeping the logic consistent. The AI doesn't have to solve every *scope* conflict upfront, it can process complex data more accurately and explain how the

different parts of a sentence connect. As the importance of machine learning continues to increase in various fields, the quality and consistency of datasets have become key issues, especially when computing power is limited. Labelling datasets has been proven to be an effective means to improve training efficiency and accuracy, providing support for improving the accuracy and reliability of data analysis.

## 2 FORMAL DEFINITION OF LINGUISTIC DATASETS

In computational linguistics, a dataset is a structured collection of linguistic data. In supervised learning, labelled target outputs of data instances are key to training models. Content labelling depends on the type of the task.

**Definition 1. Linguistic dataset** $\mathscr{D}$ is a tuple

$$\mathscr{D} = (V, P, R, L_p, L_r, S),$$

where

1. $V$ is a vocabulary, $V = \{w_1, w_2, \ldots, w_n\}$, where $w_i$ are words;

2. $P$ is a tuple of tags for parts of speech, $P = (p_1, p_2, \ldots, p_m)$, where $p_i$ represents a part of speech;

3. $R$ is a tuple of tags for binary relations, $R = (r_1, r_2, \ldots, r_k)$, where $r_i$ represents a binary relation;

4. $L_p : V \to P$ is a function for marking parts of speech;

5. $L_r : V \times V \to \mathscr{P}(R)$ is a function annotating relations;

6. $S$ is a tuple of sentences, $S = (s_1, s_2, \ldots, s_t)$.

For any sentence $s$ from $S$, we assume that

$$s = (W, E),$$

where

1. $W = (w_1, w_2, \ldots, w_n)$ is a sequence of words, $w_i \in V$;

2. $E = \{(i, j, r) \mid i, j \in [1, n], r \in \mathscr{P}(R)\}$ is a set of relation edges.

Linguistic datasets play a key role in training supervised models and can also be used in natural language processing tasks such as relation extraction. If one uses methods like machine learning to extract relationships, one needs to explain the results.

## 2.1 MATHEMATICAL REPRESENTATION OF ANNOTATED TEXT

Annotated text consists of part-of-speech tags and relations derived from the source text (the text used for model training) via categorial grammar.

Given text $T = \{w_1, w_2, \ldots, w_n\}$ where $w_i$ are words. The annotation functions are as follows:

- Part-of-speech tagging:

$$f_{\text{ps}} : T \to P, \quad P = \{p_1, p_2, \ldots, p_m\}$$

$$f_{\text{ps}}(T) = \{(w_i, p) \mid w_i \in T, p = f_{\text{ps}}(w_i)\}$$

- Relation extraction:

$$f_{\text{rel}} : T \times T \to R, \quad R = \{r_1, r_2, \ldots, r_n\}$$

$$f_{\text{rel}}(T) = \{(w_i, w_j, r) \mid w_i, w_j \in T, r = f_{\text{rel}}(w_i, w_j)\}$$

- POS encoding function:

$$g_p : P \to \mathbb{P}, \quad g_p(p_i) = \text{the } i\text{-th prime number}, \mathbb{P} \text{ is the set of prime numbers}$$

- Relation encoding function:

$$g_r : R \to \mathbb{P}, \quad g_r(r_j) = \text{the } (m+j)\text{-th prime number}$$

## 2.2 Assignment Function

Once the annotated text is obtained, parts of speech and relations between words are encoded with prime numbers, which is equivalent to assigning unique codes. Prime numbers are chosen due to the uniqueness of their products: if multiple types of relations exist between two words, they can be encoded via the product of the corresponding prime numbers. Prime encoding also ensures the interpretability of relations and grammatical categories.

1. Part-of-speech encoding function:

$$g_p : P \to \mathbb{P}, \quad g_p(p_i) = \text{the } i\text{-th prime number}$$

$$\mathbb{P} - \text{the set of prime numbers}$$

2. Relation encoding function:

$$g_r : R \to \mathbb{P}, \quad g_r(r_j) = \text{the } (m+j)\text{-th prime number}$$

**Claim 2.1.** *If the set of relations $R$ contains pairs of opposite relations, denoted as $(r, \bar{r})$, where $\bar{r}$ is the inverse relation to $r$.*

$$\psi(r) = \begin{cases} p_i & \text{if } r \text{ is a basic relation} \\ -p_i & \text{if } r = opposite(r') \text{ and } \psi(r') = p_i \end{cases}$$

*where $p_i$ is the $(m+j)$-th prime number for basic relations, and it satisfies the condition: $opposite(opposite(r)) = r$.*

Thus, not only can diverse relations be explicitly represented with a single number, but their directionality can also be indicated using positive and negative values.

## 2.3 HETEROGENEOUS GRAPH

**Definition 2.** A **heterogeneous graph** is a graph structure composed of multiple types of nodes and edges. In such graphs, nodes and edges can possess diverse attributes and relationships, representing different entities and their complex interconnections Sun & Han (2013).

In heterogeneous graphs, node types may represent distinct entity categories (e.g., users, products, topics), while edge types denote various relationships between them. Examples include "follow" relationships between users and "purchase" relationships between users and products. Both nodes and edges can carry specific attributes that characterize their properties.

Heterogeneous graphs enable simultaneous visualization of both word-to-word relationships and word-to-part-of-speech associations within a unified structure. By transforming annotated texts and datasets into heterogeneous graphs, they acquire isomorphic structures. This facilitates structural comparison between source texts and datasets through graph similarity analysis.

In this work, a heterogeneous graph $G = (V, E)$ is formally defined as:

$$G = (V_{\text{word}} \cup V_{\text{ps}}, E_{\text{ww}} \cup E_{\text{wp}})$$

where:

- $V_{\text{word}} = \{w_1, w_2, \ldots, w_n\}$ represents word nodes.
- $V_{\text{ps}} = \{p_1, p_2, \ldots, p_m\}$ represents part-of-speech nodes.
- $E_{\text{ww}}$ represents word-word relations.
- $E_{\text{wp}}$ represents word-POS relations.

For Annotated Texts, given an input text $T$ with its annotations, the graph $G$ is built through:

- Adding word nodes:
$$V_{\text{word}} := V_{\text{word}} \cup \{w_i\}, w_i \in T$$

- Adding POS nodes:
$$V_{\text{ps}} := V_{\text{ps}} \cup \{p_i\}, p_i \in P - \text{the set of parts of speech}$$

- Adding POS edges:
$$E_{\text{wp}} := E_{\text{wp}} \cup \{(w_i, p_i, g_p(p_i))\}$$

- Adding relation edges:
$$E_{\text{ww}} := E_{\text{ww}} \cup \left\{ \left( w_i, w_j, \prod_{r \in f_{\text{rel}}(w_i, w_j)} g_r(r) \right) \right\}$$

Graphs $G_1$ from dataset $D_1$ and $G_2$ from dataset $D_2$ are constructed using the same method. Taking $G_1 = (V_1, E_1)$ as an example. For dataset $D_1 = (V_1, P_1, R_1, L_{p1}, L_{r1}, S_1)$:

- Adding word nodes:
$$V_{\text{word}}^1 = \{w_1, w_2, \ldots, w_n\} = V_1 \in D_1$$

- Adding POS nodes:
$$V_{\text{ps}}^1 = \{p_1, p_2, \ldots, p_m\} = P_1 \in D_1$$

- Adding POS edges:
$$E_{\text{wp}}^1 = \{(w, p, g_p(p)) \mid w \in V_{\text{word}}^1, p = L_{p1}(w)\}$$

- Adding relation edges:
$$E_{\text{ww}}^1 = \left\{ \left( w_i, w_j, \prod_{r \in L_{r1}(w_i, w_j)} g_r(r) \right) \mid L_{r1}(w_i, w_j) \neq \emptyset \right\}$$

## 2.4 GRAPH EDIT DISTANCE (GED)

**Definition 3.** The **Graph Edit Distance (GED)** between two graphs $G_1$ and $G_2$ is defined as the minimum cost of edit operations required to transform $G_1$ into a graph isomorphic to $G_2$ Sanfeliu & Fu (1983):

$$\text{GED}(G_1, G_2) = \min_{(e_1, \ldots, e_k) \in \mathscr{P}(G_1, G_2)} \sum_{i=1}^{k} c(e_i)$$

where:

$\mathscr{P}(G_1, G_2)$ denotes the set of all valid edit paths transforming $G_1$ into a graph isomorphic to $G_2$, edit path consists of a sequence of node or edge operations.

$c(e_i) \geq 0$ represents the cost associated with edit operation $e_i$.

Taking graph $G$ which from annotated text as a basis, we modify graphs $G_1$ and $G_2$ in accordance with graph $G$ and calculate the required cost $\text{GED}(G_1, G) \in \mathbb{N}$ and $\text{GED}(G_2, G) \in \mathbb{N}$.

$$\text{GED}(G_i, G) = C_{\text{node}}^{(i)} + C_{\text{edge}}^{(i)} \quad \text{for } i \in \{1, 2\}$$

Node Operations Cost:

$$C_{\text{node}}^{(i)} = \sum_{v \in V_i \setminus V} c_{\text{del}}(v) + \sum_{v \in V \setminus V_i} c_{\text{add}}(v), \quad \text{where} \quad c_{\text{del}}(v) = c_{\text{add}}(v) = 1$$

- Deletion $c_{\text{del}}(v)$: Remove node $v \notin V$ from $V_i$
- Addition $c_{\text{add}}(v)$: Insert node $v \in V$ into $V_i$

Edge Operations Cost:

$$C_{\text{edge}}^{(i)} = \sum_{e \in E_i \setminus E} c_{\text{del}}(e) + \sum_{e \in E \setminus E_i} c_{\text{add}}(e), \quad \text{where} \quad c_{\text{del}}(e) = c_{\text{add}}(e) = 1$$

- The edge $e = (u, v, w)$ is the edge from node $u$ to $v$ whose value is $w$.
- Deletion $c_{\text{del}}(e)$: Remove edge $e \in E_i$ when $\nexists e' \in E$ has the same endpoints and value
- Addition $c_{\text{add}}(e)$: Insert edge $e \in E$ when $\nexists e' \in E_i$ has the same endpoints and value

The **Normalized Graph Edit Distance (NormGED)** quantifies the structural divergence. It allows for a fair comparison of models when their graphs are of different sizes.

$$\text{NormGED}(G_i, G) = \frac{\text{GED}(G_i, G)}{|V_i| + |E_i| + |V| + |E|} \in [0, 1]$$

where:

$$\begin{cases} 0: & \text{Perfect alignment (no edit operations needed)} \\ 1: & \text{Complete divergence (every node/edge requires operations)} \end{cases}$$

Lower NormGED values indicate better model performance.

## 2.5 EXAMPLE

Here is a simple example based on the sentence: **"People like cats and dogs as pets".**

Using methods such as machine learning, two different results are obtained:

- $D_1$: "people", "cats", "dogs", "pets" — nouns; "like" — verb; relations (people,cats), (people,dogs), (people,pets) — "like".
- $D_2$: "people", "cats", "dogs", "pets" — nouns; "like" — verb, "as" — preposition; relations (people,cats) — "like", (dogs,pets) — "as".

Analysis of the sentence according to combinatory categorial grammar (Table 1):

| Word | Syntactic Category | $\lambda$-term |
|------|--------------------|----------------|
| People | NP | `people` |
| like | $(NP\backslash S)/PP/NP$ | $\lambda x.\lambda y.\lambda p.\,\text{like}(y,x) \wedge \text{as}(x,p)$ |
| cats | NP | `cats` |
| and | $(NP\backslash NP)/NP$ | $\lambda a.\lambda b.\,\text{and}(b,a)$ |
| dogs | NP | `dogs` |
| as | $PP/NP$ | $\lambda p.\lambda x.\,\text{as}(x,p)$ |
| pets | NP | `pets` |

**Table 1:** *CCG analysis of example sentence.*

According to the syntax, it can be concluded that type NP is a noun, type $(NP\backslash S)/PP/NP$ is a verb, type $(NP\backslash NP)/NP$ is a conjunction, and type $PP/NP$ is a preposition. Based on the combination of semantics, it can be concluded that the relation between "people" and "cats", "dogs" is "like", while the relation between "cats", "dogs" and "pets" is "as".

Parts of speech and relations are encoded using prime numbers, as shown in the (Table 2):

| Part of Speech and Relation | Prime Number Code |
|------------------------------|-------------------|
| Noun | 2 |
| Verb | 3 |
| Conjunction | 5 |
| Preposition | 7 |
| like | 11 |
| as | 13 |

**Table 2:** *Prime number encoding for parts of speech and relations.*

Based on the parts of speech of words and their corresponding part-of-speech codes, as well as the relations between words and their corresponding relation codes, a heterogeneous graph $G$ of the source text can be constructed:

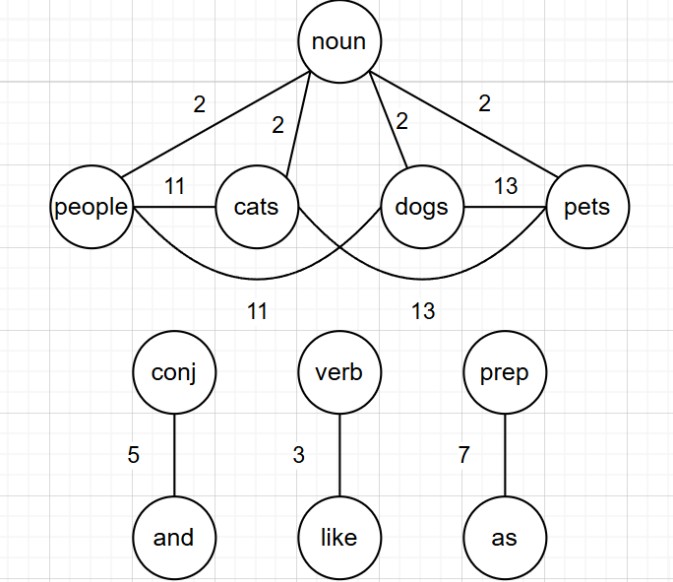

**Figure 1:** *Ground truth heterogeneous graph G from annotated text.*

The encoding values shown in the table are also used for parts of speech and relations in datasets $D_1$ and $D_2$. If an unknown part of speech or relation appears, a new prime number code is added.

Similarly, heterogeneous graphs $G_1$ and $G_2$ for $D_1$ and $D_2$ are obtained:

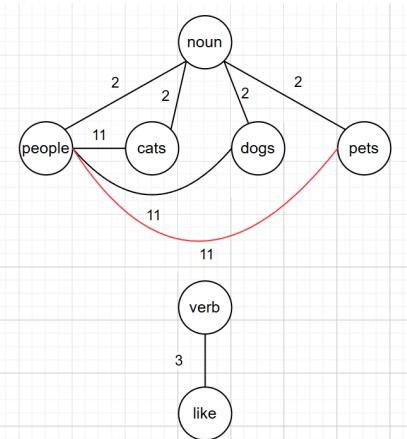

**Figure 2:** *Heterogeneous graph $G_1$ for dataset $D_1$.*

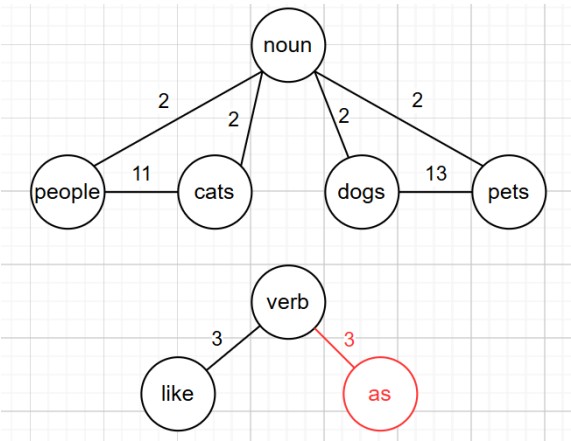

**Figure 3:** *Heterogeneous graph $G_2$ for dataset $D_2$.*

Using the annotated text graph $G$ to represent the true information, measure the graphs of datasets $G_1$ and $G_2$:

1. Calculate the graph edit distance from graph $G_1$ to $G$:

$$
\begin{aligned}
\mathrm{GED}(G_1, G) &= C_{\mathrm{node}} + C_{\mathrm{edge}} \\
&= C_{\mathrm{add\_node}} + C_{\mathrm{del\_edge}} + C_{\mathrm{add\_edge}} \\
&\quad \text{conj, and, prep, as (people,pets) (cats,pets),(dogs,pets),(conj,and),(perp,as)} \\
&= 9
\end{aligned}
$$

$$
\begin{aligned}
\mathrm{NormGED} &= \frac{9}{7 + 8 + 11 + 11} \\
&= 0.243
\end{aligned}
$$

2. Similarly, the graph edit distance from graph $G_2$ to $G$:

$$
\begin{aligned}
\mathrm{GED}(G_2, G) &= C_{\mathrm{node}} + C_{\mathrm{edge}} \\
&= C_{\mathrm{add\_node}} + C_{\mathrm{del\_edge}} + C_{\mathrm{add\_edge}} \\
&\quad \text{conj, and, prep (as,verb) (people,dogs),(cats,pets),(conj,and),(perp,as)} \\
&= 8
\end{aligned}
$$

$$
\begin{aligned}
\mathrm{NormGED}' &= \frac{8}{8 + 8 + 11 + 11} \\
&= 0.210
\end{aligned}
$$

Since $\mathrm{NormGED} > \mathrm{NormGED}'$, $G_2$ (corresponding to dataset $D_2$) is better.

The graph edit distance methodology described above measures structural similarity between linguistic datasets. However, structural comparison alone cannot resolve semantic ambiguities arising from quanti-

fier scope. To address this limitation, we integrate formal logical methods that complement our graph-based approach. The heterogeneous graphs constructed in Section 2.3 can be extended with $\varepsilon$-terms and Skolem functions as additional node types, enabling unambiguous semantic representation while preserving the graph structure suitable for GED computation.

## 2.6 Formal Methods for Resolving Quantifier Scope Ambiguity in AI Systems

One key challenge in natural language processing (NLP) and automated theorem proving is quantifier scope ambiguity. This arises when a sentence's logical structure allows multiple interpretations due to different quantifier orders.

A classic example is "Every student read some book", which can mean either "all students read the same book" or "each student read their own book". Formally, these are:

$$\exists y \, (B(y) \wedge \forall x \, (S(x) \rightarrow R(x,y))) \tag{1}$$

(one book for all students) and

$$\forall x \, (S(x) \rightarrow \exists y \, (B(y) \wedge R(x,y))) \tag{2}$$

(each student their own book),

where $S(x)$: "$x$ is a student". $B(y)$: "$y$ is a book", $R(x,y)$: "$x$ read $y$."

Several methods address this issue, such as type shifting or quantifier raising. However, these often alter syntax and semantics in complex ways. This work focuses on two effective formal approaches: $\varepsilon$-calculus and skolem functions Heusinger (2004).

### 2.6.1 Integration of $\varepsilon$-Terms into Heterogeneous Graphs

To integrate epsilon-terms into our graph framework, we extend the heterogeneous graph definition from Section 2.3.

**Definition 4** ($\varepsilon$-Extended Heterogeneous Graph). Given a heterogeneous graph $G = (V_{\text{word}} \cup V_{\text{ps}}, E_{\text{ww}} \cup E_{\text{wp}})$, the $\varepsilon$-**extended graph** $G^{\varepsilon}$ is defined as:

$$G^{\varepsilon} = (V_{\text{word}} \cup V_{\text{ps}} \cup V_{\varepsilon}, E_{\text{ww}} \cup E_{\text{wp}} \cup E_{\varepsilon})$$

where:

- $V_{\varepsilon} = \{v_{\varepsilon,P} \mid \varepsilon y.P(y)$ appears in semantic representation$\}$ represents epsilon-choice nodes

- $E_{\varepsilon} = \{(v_{\varepsilon,P}, v_x, \text{choice}) \mid x$ satisfies predicate $P\}$ represents choice edges

For the sentence "Every student read some book" with narrow scope interpretation:

$$\forall x \, (S(x) \rightarrow R \, (x, \varepsilon y \, (B(y) \wedge R(x,y))))$$

the graph $G^{\varepsilon}$ contains:

- Node $v_{\varepsilon, B \wedge R} \in V_{\varepsilon}$ representing the choice of a book
- Edges $(v_{\varepsilon, B \wedge R}, v_{\text{book}_i}, \text{choice})$ for each book candidate
- Edges $(v_{\text{student}_j}, v_{\varepsilon, B \wedge R}, \text{depends})$ showing functional dependency

This approach uses $\varepsilon$-**terms** to select a specific object satisfying a predicate. Formally, $\varepsilon y P(y)$ denotes a $y$ where $P(y)$ holds Zach (2016).

For the second interpretation Hetzl (2024):

$$\forall x \left( S(x) \rightarrow R\left(x, \varepsilon y \left(B(y) \wedge R(x,y)\right)\right)\right) \tag{3}$$

The $\varepsilon$-term picks a book *for each* student, ensuring compositional structure and interpretability—key for NLP tasks like semantic parsing and anaphora resolution in chatbots Chatzikyriakidis et al. (2017).

**Impact on Graph Edit Distance.** Different scope interpretations produce different $\varepsilon$-extended graphs:

- **Wide scope** ($\exists y \forall x$): single $\varepsilon$-node connected to all student nodes
- **Narrow scope** ($\forall x \exists y$): multiple $\varepsilon$-nodes (one per student)

The NormGED between these graphs quantifies the structural impact of scope ambiguity resolution, providing a measurable criterion for choosing between interpretations.

### 2.6.2 SKOLEM FUNCTIONS

Skolemization eliminates existential quantifiers by replacing them with functions depending on free variables. For $\forall x \exists y R(x,y)$, introduce $f$ such that:

$$\forall x R(x, f(x)) \tag{4}$$

This simplifies formulas, reduces search space in theorem provers, and aids NLP logic generation.

In the graph framework, Skolem functions are represented as:

- Function nodes $v_f \in V_{\text{func}}$ added to the graph
- Edges $(v_x, v_f, \text{argument})$ and $(v_f, v_{f(x)}, \text{result})$

Both methods boost AI accuracy. $\varepsilon$-calculus offers flexibility for NLP dependencies; Skolem functions simplify proofs and model checking. They excel in hybrid logic-neural systems but require managing computational costs in large-scale applications Mints (1996).

## 2.7 INTEGRATION SUMMARY

The proposed framework combines three complementary approaches:

1. Graph-based representation (Sections 2.3–2.5) for structural comparison of linguistic datasets via NormGED;
2. Prime encoding (Section 2.2) for interpretable relation representation;
3. Formal logic methods (Section 2.6) for resolving semantic ambiguities that affect graph structure.

Together, these methods enable explainable AI systems that can both measure dataset similarity and maintain logical consistency in semantic representations. Future work will implement the integration of $\varepsilon$-terms directly into the heterogeneous graph construction algorithm.

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
