# OpenReview forum: "Linguistic Structures for Formal Semantics of Natural Languages"
_mathai.club/MathAI/2026/Conference — 2026 Oral_

### Official Review · Reviewer_GRR7 · 2026-03-11
**A formally rigorous framework for representing and comparing linguistic datasets, with interesting algebraic encoding; requires stronger empirical grounding and broader contextualization.**

**Rating:** 7
**Confidence:** 4

**Review:**

The paper presents a formal mathematical framework for representing linguistic datasets as heterogeneous graphs, measuring their structural similarity via Graph Edit Distance (GED), and resolving quantifier scope ambiguity using Hilbert's ε-operator and Skolem choice functions.

**Strengths:**

- The use of prime number encoding for part-of-speech tags and relational labels is an elegant and original contribution. The fundamental theorem of arithmetic guarantees unique factorization, making composite edge weights interpretable and invertible — a property not commonly exploited in graph-based NLP representations.
- The extension of heterogeneous graphs with ε-nodes and Skolem function nodes to represent scope ambiguity is a well-motivated formal integration. The resulting framework unifies structural comparison (NormGED) with logical disambiguation in a single representational layer.
- The formal definitions are stated precisely and the worked example (Section 2.5) is helpful for understanding the GED computation.

**Weaknesses and questions to the authors:**

1. **Bootstrapping of annotations.** The entire pipeline — graph construction, GED computation, and ε-term integration — presupposes that POS tags and inter-word relations are already correctly annotated. However, obtaining such annotations is itself a non-trivial task, often requiring the same kind of disambiguation the framework aims to resolve. Could the authors clarify whether the proposed framework is intended as a post-processing step over existing NLP pipelines, or whether it is self-contained? What is the sensitivity of NormGED to annotation noise or inconsistency?

2. **Computational complexity.** GED is known to be NP-hard in general. The paper's setup — comparing dataset graphs $G_1$, $G_2$ against a fixed ground-truth graph $G$ derived from annotated text — potentially sidesteps the full generality of the problem, since one operand is fixed and the graph structure is linguistically constrained. This is actually a promising aspect of the approach, but it is not discussed. The authors are encouraged to analyze whether this anchoring to a reference graph admits a more efficient algorithm, and if so, to make this a stated contribution rather than leaving the complexity question open.

3. **Scope of evaluation.** The paper provides a single illustrative example. It would substantially strengthen the work to include even a small-scale quantitative evaluation — e.g., correlation of NormGED with human similarity judgments, or performance on a standard scope disambiguation benchmark.

4. **Relation to prior work.** The bibliography contains only six entries, which makes it difficult to assess the novelty and positioning of the contribution.

5. **Limitations section.** The paper does not discuss limitations of the approach — for instance, scalability to long documents, handling of languages with free word order, or the assumption that the ground-truth annotated graph $G$ is available for NormGED computation.

**Minor:** The statement that formulas (1) and (3) in Section 2.6.1 are both labeled as the narrow scope interpretation appears to be a typographical error — the wide scope reading is not explicitly given in the ε-notation, which may confuse readers.

---

## Conclusion

The paper introduces a formally coherent and algebraically interesting representation framework. The prime encoding scheme and the graph-based integration of ε-calculus are genuine contributions worth publishing. The main gap is the absence of empirical validation and insufficient engagement with related literature. The authors are encouraged to expand the experimental section, add a limitations discussion, and significantly extend the bibliography before final submission.

---

### Official Review · Reviewer_ydM3 · 2026-03-12
**Linguistic Structures for Formal Semantics of Natural Languages**

**Rating:** 9
**Confidence:** 5

**Review:**

This paper proposes a mathematical framework for formalizing natural language semantics, combining three components:
1. Graph representation: using heterogeneous graphs to visualize relationships between words and parts of speech (pp. 3-4).
2. Comparison algorithm (NormGED): using normalized graph edit distance (Graph Edit Distance) to assess the similarity between labeled datasets and the "gold standard" (annotated text) (pp. 5, 8).
3. Logical methods: integrating the Hilbert calculus and Skolem functions to solve the scope problem (pp. 9-10).
Novelty Assessment
• An elegant way to represent multi-representational relationships in a single numerical value (p. 3).
Applied Relevance for XAI: The proposed model is oriented toward "Explainable AI," enabling not only training models but also mathematically justifying differences in text interpretations through visualized graph structures (pp. 1, 10). The work has a moderately high scientific novelty, which is manifested in the following aspects:
• Hybrid Approach: The novelty lies not in the methods themselves   but in their synthesis. The authors propose extending classical GED with semantic nodes, which allows for quantitative measurement of the influence of logical interpretation on sentence structure (pp. 9-10).
• Strengths: Clear mathematical formalization (Definitions 1–4) and a practical example of calculating NormGED for two different interpretations of the same sentence (pp. 6, 8).
• Weaknesses: The current version lacks empirical testing on large datasets.
Conclusion:This article presents a qualitative theoretical study at the intersection of mathematical linguistics and graph theory, proposing a specific toolkit for formal verification of semantic models.

---

### Official Review · Reviewer_cJzk · 2026-03-13
**This paper introduces a formal, graph-based framework for representing, comparing, and analyzing linguistic datasets. The core contribution is a method to transform annotated text (with parts of speech and relations) into heterogeneous graphs using a novel prime number encoding scheme for linguistic tags. The paper then proposes using the Normalized Graph Edit Distance (NormGED) to quantify the structural similarity between a "ground truth" graph (from an annotated source text) and the graphs produced by different NLP models or datasets. Furthermore, it outlines a theoretical integration of formal logic methods (epsilon-calculus and Skolem functions) to resolve semantic ambiguities like quantifier scope, extending the graph structure to accommodate these logical constructs. The primary strengths lie in the novelty of combining graph theory with prime encoding for interpretability and the ambitious goal of bridging structural and semantic analysis. However, the paper in its current form is severely limited by lack of experimental validation, and a disconnect between the proposed framework and a demonstration of its practical utility. While the mathematical foundation is sound, the work reads more like a promising research proposal than a finished, empirically validated study.**

**Rating:** 5
**Confidence:** 4

**Review:**

### Brief Summary

This paper introduces a formal, graph-based framework for representing, comparing, and analyzing linguistic datasets. The core contribution is a method to transform annotated text (with parts of speech and relations) into heterogeneous graphs using a novel prime number encoding scheme for linguistic tags. The paper then proposes using the Normalized Graph Edit Distance (NormGED) to quantify the structural similarity between a "ground truth" graph (from an annotated source text) and the graphs produced by different NLP models or datasets. Furthermore, it outlines a theoretical integration of formal logic methods (epsilon-calculus and Skolem functions) to resolve semantic ambiguities like quantifier scope, extending the graph structure to accommodate these logical constructs. The primary strengths lie in the novelty of combining graph theory with prime encoding for interpretability and the ambitious goal of bridging structural and semantic analysis. However, the paper in its current form is severely limited by lack of experimental validation, and a disconnect between the proposed framework and a demonstration of its practical utility. While the mathematical foundation is sound, the work reads more like a promising research proposal than a finished, empirically validated study.

### Detailed Review

**Overview:**
The paper addresses the problem of how to formally represent, compare, and evaluate linguistic datasets, particularly in the context of explainable AI for NLP tasks like relation extraction. The central idea is to convert linguistic annotations into a structured, mathematical object -- a heterogeneous graph -- that can be manipulated and compared. The paper proceeds by:

1.  Formally defining a linguistic dataset.
2.  Introducing a prime-number encoding for parts of speech and relations to ensure unique, interpretable, and composable representations.
3.  Defining a method to construct a heterogeneous graph from this encoded data, with distinct node types for words and parts-of-speech, and edges for word-word relations and word-POS assignments.
4.  Proposing the use of Graph Edit Distance (GED), normalized, as a metric to compare the structural fidelity of a model's output graph against a ground-truth graph.
5.  Discussing the limitations of purely structural comparison (e.g., quantifier scope ambiguity) and suggesting an extension to the graph model using epsilon-terms and Skolem functions to capture logical semantics.

**Strengths:**

1.  The combination of heterogeneous graphs and prime number encoding for linguistic features is creative. Using the product of primes to represent multiple relations between two words is an elegant solution that maintains interpretability and uniqueness.
2.  The paper provides clear, mathematical definitions for its core concepts (Dataset, Heterogeneous Graph, GED). This rigor is essential for a methodology paper and leaves little room for ambiguity in the proposed framework.
3.  The authors correctly identify a key weakness of purely structural methods -- their inability to handle semantic nuances like quantifier scope. The attempt to bridge this gap by integrating epsilon-calculus and Skolem functions into the graph framework is ambitious and theoretically interesting, pointing towards a more robust and explainable AI system.
4.  The step-by-step example using the sentence "People like cats and dogs as pets" is helpful. It grounds the abstract definitions and demonstrates, in principle, how the graph construction and GED calculation would work.

**Weaknesses:**

1.  The paper is entirely theoretical. It proposes a metric (NormGED) to determine which dataset is "better" but provides no experimental results to validate this claim. Does a lower NormGED correlate with better performance on a downstream task like relation extraction? Is this metric stable across different text genres? Without experiments on real-world datasets, the utility of the entire framework remains unproven.
2.  While elegant, encoding a potentially large set of relations and parts-of-speech with prime numbers is computationally problematic. The product of several primes can grow astronomically large, quickly exceeding the limits of standard integer representations. The paper does not address this scalability issue or discuss potential workarounds (e.g., using big integers or hashing strategies that retain the multiplicative property).
3.  Sections 2.6.1 and 2.6.2 introduce epsilon-terms and Skolem functions but the integration feels tacked on. It describes how they could be added to the graph as new node types, but it doesn't explain how this would change the GED calculation or how the "choice" edges would be weighted or compared. It states the problem and a high-level solution but lacks the formal depth present in the earlier sections.
4.  While formally rigorous, the paper jumps between concepts (datasets, annotated text, graphs) without a clear, high-level narrative flow. For instance, the relationship between a "linguistic dataset" (Definition 1) and an "annotated text" (Section 2.1) could be explained more explicitly. The sudden introduction of CCG in the example, without prior explanation, may also confuse readers not deeply familiar with it.

**Suggestions for Improvement:**

1.  **Add Empirical Evaluation:** This is critical. The authors must implement their framework and test it on standard NLP benchmarks. For example, they could take a dataset like Penn Treebank or a relation extraction corpus, create ground-truth graphs, generate outputs from several models (e.g., a rule-based system, an LSTM, a Transformer), compute NormGED for each, and then analyze if NormGED scores correlate with traditional performance metrics (F1-score, accuracy). This would validate the metric's usefulness.
2.  **Address Computational Complexity:** Discuss the scalability of the prime encoding method. Acknowledge the potential for integer overflow and propose a solution (e.g., using arbitrary-precision arithmetic or a factorization-based approach). A complexity analysis of the GED calculation on these graphs should also be included, as GED is known to be NP-hard, even if the proposed graphs are small.
3.  **Deepen the Integration of Formal Logic:** Instead of just stating the possibility, provide a concrete, worked example for quantifier scope (like "Every student read some book") that shows the full graph construction, the GED calculation between the two scope interpretations, and how the metric would inform a choice. This would demonstrate the power of the combined approach.
4.  **Improve Narrative Flow:** Restructure the paper to tell a clearer story. Start with the problem: "How can we compare linguistic datasets in an interpretable way?" Then introduce graphs as the solution, prime encoding as a key mechanism, and GED as the comparison tool. Finally, present the semantic ambiguity problem as a limitation of this structural view and introduce the formal logic extensions as a way to overcome it. The example should be woven into this narrative.

---

### Decision · Program_Chairs · 2026-03-14

**Decision:**

Accept (Oral)

**Comment:**

Dear Author(s),

On behalf of the Program Committee of the International Conference on Mathematics of Artificial Intelligence (MathAI 2026), we are pleased to inform you that your paper has been accepted for an oral presentation at MathAI 2026.

Your paper was evaluated through a rigorous two-stage review process involving both automated screening and expert review by members of the Program Committee. The reviewers recognized the quality and contribution of your work.

Presentation details:

- Format: Oral presentation (15–20 minutes + 5 minutes Q&A)
- Mode: You may present either in person (offline) at the conference venue in Sirius, Russia, or remotely via Zoom. Please indicate your preferred mode when confirming your participation.
- Conference dates: Marh 30 - April 3, 2026
- Website: https://mathai.club

Next steps:

1. Please confirm your participation and presentation mode by replying to this email mathai.club@yandex.ru no later than March 15, 2026 18:00 Moscow time.
2. If you plan to attend in person, the organizing committee will provide accommodation details separately.
3. Please prepare your final camera-ready manuscript according to the formatting guidelines available at https://mathai.club and upload it to OpenReview by March 15, 2026 18:00 Moscow time.

Should you have any questions regarding the program, logistics, or your presentation slot, please do not hesitate to contact us.

We look forward to your contribution to MathAI 2026.

With kind regards,

MathAI 2026 Program Committee
International Conference on Mathematics of Artificial Intelligence
https://mathai.club
OpenReview: https://openreview.net/group?id=mathai.club/MathAI/2026/Conference
Telegram: https://t.me/MathAI_club
Email: mathai.club@yandex.ru